Broad-scale spatial distribution, microhabitat association and habitat partitioning of damselfishes (family Pomacentridae) on an Okinawan coral reef

Nanami Atsushi nanami@fra.affrc.go.jp
Yaeyema Field Station, Fisheries Technology Institute, Japan Fisheries Research and Education Agency , Ishigaki, Okinawa , Japan
Manjarrez Javier
Electronic publication date: 2025 Feb 14
Publication date: 2025
Volume: 13
Electronic Location ID: e18977
Received 2024 Sep 19; Accepted 2025 Jan 22
Copyright: © 2025 Nanami
Copyright year: 2025
Copyright holder: Nanami
License: This is an open access article distributed under the terms of the Creative Commons Attribution License, which permits unrestricted use, distribution, reproduction and adaptation in any medium and for any purpose provided that it is properly attributed. For attribution, the original author(s), title, publication source (PeerJ) and either DOI or URL of the article must be cited.
License URL: https://creativecommons.org/licenses/by/4.0/

Keywords: Broad-scale spatial distribution, Microhabitat association, Habitat partitioning, Damselfish, Substrate characteristics, Coral reef

Funding: Environment Research and Technology Development S-15-3(4): JPMEERF16S11513 Ministry of the Environment, Japan This study was funded by the Environment Research and Technology Development Fund (S-15-3(4): JPMEERF16S11513) of the Ministry of the Environment, Japan. The funders had no role in study design, data collection and analysis, decision to publish, or preparation of the manuscript.

==============================
Spatial distribution of coral reef fishes is related to diverse environmental variables. This study aimed to elucidate the (1) broad-scale spatial distribution, (2) microhabitat-scale substrate association, (3) degree of dependence on live corals and (4) habitat partitioning of 26 damselfish species on an Okinawan coral reef. Broad-scale analysis revealed that fish assemblages could be divided into three groups in relation to the degree of wave exposure, and the coverage of live corals as well as non-coralline substrates: (1) 11 species that were found in exposed reefs with greater coverage of rock; (2) nine species that that were found in inner reefs with greater coverage of live corals, dead corals and macroalgae; and (3) six species that were found in inner reefs with a greater coverage of sand. Microhabitat-scale analysis revealed that fish assemblages could be divided into six groups in relation to diverse microhabitat availability: (1) 12 species showed significant positive associations with rock; (2) two species showed significant positive associations with coral rubble; (3) two species showed significant positive associations with corymbose Acropora, Pocillopora and branching corals; (4) three species showed significant positive associations with branching Acropora; (5) three species showed significant positive associations with branching Acropora, branching Isopora and branching Porites; and (6) two species showed significant positive associations with staghorn Acropora and branching Millepora. The microhabitat-scale analysis also revealed that Pomacentrus amboinensis showed a significant positive association with branching Millepora, whereas Neopomacentrus anabatoides showed significant positive associations with branching Porites, foliose coral and dead branching Porites. Among the 26 species, nine species were categorized as obligate coral dwellers (>80% of the individuals were associated with live corals), and three species showed a greater degree of dependence on acroporid corals (>60% individuals were associated with acroporid coral). Niche overlap analysis revealed that 14 species showed relatively greater habitat partitioning with other species, whereas the remaining 12 species showed greater habitat overlaps among some species. These results suggest that broad-scale and microhabitat-scale habitat partitioning is one of the factors supporting coexistence in at least 14 species among the 26 species, and the effects of habitat diversity on the species coexistence of damselfishes should be incorporated to establish effective marine protected areas to preserve damselfish species diversity.

Introduction

Coral reefs support diverse species of marine organisms. Among these, coral reef fishes show close relationships between species-specific spatial distributions and diverse environmental characteristics provided by live corals, non-coralline substrates, and topographic complexity (Luckhurst & Luckhurst, 1978; Williams, 1991; Friedlander et al., 2003; Emslie et al., 2010). A combination of broad-scale and microhabitat-scale approaches provides a more comprehensive understanding of these relationships (Syms, 1995; Nanami, 2023). Considering that a single approach might overlook some ecological factors that other approaches might detect, integrating two spatial-scale approaches could provide a more comprehensive understanding of fish spatial distribution in relation to environmental characteristics (Syms, 1995; Eagle, Jones & McCormick, 2001; Nanami, 2023). Thus, this study combined broad-scale and microhabitat-scale approaches to enable a more comprehensive understanding of the relationship between fish and environmental characteristics.

Broad-scale approaches provide species-specific spatial distribution of fishes at a scale of several kilometers or several tens of kilometers. In this approach, large-scale environmental variables, such as topographic characteristics (reef slope, reef crest, and reef flat), water depth, and degree of wave exposure, have marked effects on the spatial distribution of various fish families (Fulton, Bellwood & Wainwright, 2001; Friedlander et al., 2003; Hoey & Bellwood, 2008; Pratchett & Berumen, 2008; Emslie et al., 2010; Cheal et al., 2012; Goatley, González-Cabello & Bellwood, 2016; Nanami, 2018, 2020; Benthuysen et al., 2022).

By contrast, microhabitat-scale approaches provide species-specific habitat associations within a scale of several centimeters and treat fine scale variables such as species of live corals and forms of non-coralline substrates. Many fish families exhibit diverse species-specific microhabitat associations with various substrates (Syms, 1995; Munday, Jones & Caley, 1997; Munday, 2004; Doll et al., 2021). These studies have also shown that fishes can be primarily divided into two categories; specialists (species showing greater habitat specialization to particular substrates) and generalists (species showing a broader extent of habitat selection for various substrates). For example, Pratchett et al. (2016) showed that damselfishes can be divided into obligate and facultative coral dwellers, in which obligate coral dwellers are defined as species in which over 80% of the individuals are associated with live corals. Clarifying the degree of dependence on live corals is important, since such information can be applied to select candidate sites for protection or habitat restoration of coral reef fishes. This is because live corals, especially acroporid corals, have shown population declines due to global climate change (Marshall & Baird, 2000; McClanahan et al., 2004), and such a decline of the coral population can cause significantly negative impacts on coral reef fish populations (Wilson et al., 2006; Pratchett et al., 2008).

Another ecological aspect of species diversity is the examination of the factors supporting the coexistence among diverse species (Tokeshi, 1999; Albrecht & Gotelli, 2001; Darmon et al., 2012; Harmácková, Remesová & Remes, 2019; Salas-López et al., 2022), and a greater degree of habitat partitioning has been shown to be a main factor in maintaining species coexistence in coral reef fishes at broad-scale and microhabitat-scale. Broad-scale surveys have revealed that distance from the mainland (near-shore, intermediate shore and offshore), topographic gradients (reef slope, reef crest and reef flat) and depth gradients affected the habitat partitioning among acanthurids, chaetodontids, gobiids and pomacentrids (Clarke, 1977; Robertson & Lassig, 1980; Williams, 1982, 1991; Bouchon-Navaro, 1986; Goatley, González-Cabello & Bellwood, 2016). In contrast, microhabitat-scale surveys have shown that coral morphology, coral species and various types of non-coralline substrates support habitat partitioning among apogonids and gobiids (Munday, Jones & Caley, 1997; Gardiner & Jones, 2005; Doll et al., 2021). These studies suggest that a greater degree of habitat partitioning allows species coexistence of coral reef fishes, and a multiple spatial scale approach is useful for a more comprehensive understanding of habitat partitioning among multiple coral reef fish species.

Clarifying the precise spatial distribution of target fish species in relation to substrate characteristics is also important when considering the location of marine protected areas (MPAs), which have been implemented to effectively conserve coral reef fishes. If the effects of habitat diversity on the species coexistence of coral reef fishes can be clarified by the integration of broad- and microhabitat-scale approaches, then effective MPAs that help to preserve the species diversity of coral reef fishes would be established.

Damselfishes (family Pomacentridae) are a major fish group that consists of diverse species in coral reefs (Wilson et al., 2006, 2008; Pratchett et al., 2012, 2016). Considering that damselfishes consist of diverse species, this fish group provides an ideal study system for clarifying the ecological mechanism underlying species coexistence. In clarifying the ecological mechanism, spatial distribution and microhabitat associations should be examined. Previous studies have shown broad-scale spatial variation in damselfish assemblages in the Great Barrier Reef (Meekan, Steven & Fortin, 1995; Bay, Jones & McCormick, 2001; Chaves et al., 2012; Eurich, McCormick & Jones, 2018a; Emslie, Logan & Cheal, 2019; Emslie et al., 2012). Microhabitat-scale substrate associations in damselfishes have also been reported (Waldner & Robertson, 1980; Wilson et al., 2008; Nadler et al., 2014; Komyakova, Munday & Jones, 2019). In addition, species-specific differences in responses to habitat loss caused by the degradation of coral assemblages have been suggested (Pratchett et al., 2012, 2016; Wilson et al., 2006, 2008). Clarification of microhabitat associations of damselfishes would provide useful information for identifying potential MPAs and for the restoration of specific substrates to conserve the population and species diversity of damselfishes.

In Okinawan coral reefs, Nanami & Nishihira (2002) and Nanami et al. (2005) revealed a broad-scale difference in the assemblage structure of damselfishes in relation to water depth and the degree of wave exposure. Nanami & Nishihira (2003) also showed microhabitat associations in six damselfish species. However, the spatial distribution of damselfishes by integrating with multiple spatial scales and the degree of dependence on live corals as habitats have not yet been examined in this region. In addition, the degree of habitat partitioning that supports coexistence among damselfish species remains unclear. Thus, this study aimed to clarify the (1) broad-scale species-specific spatial distribution in relation to substrate characteristics, (2) microhabitat-scale substrate association in relation to substrate availability, (3) degree of dependence on live corals, and (4) degree of habitat partitioning from the two spatial scale approaches among damselfish species in an Okinawan coral reef. The results obtained from the two spatial scale approaches enable a more comprehensive understanding of the spatial distribution in relation to the substrate characteristics and the mechanisms of species coexistence in damselfishes.

Materials and Methods

Broad-scale fish spatial distribution

Underwater visual surveys were conducted at Sekisei Lagoon and Nagura Bay on the Yaeyama Islands, Okinawa, Japan from July to December 2019 (Figs. 1A, 1B). A total of 67 study sites (31 sites on exposed reefs and 36 sites on inner reefs) were established with an inter-site distance of ~2 km (Fig. 1C).

Figure 1 Maps showing the location of the Yaeyama Islands (A), study area (B), 67 study sites for examining broad-scale spatial distributions (C), and 19 sites for examining microhabitat associations (D).

In (C), magenta and yellow symbols represent the sites in the exposed reefs and inner reefs, respectively. (A) Map created by processing Geospatial Information Authority (https://mapps.gsi.go.jp/maplibSearch.do#1). The aerial photographs used in (B), (C) and (D) were provided by the International Coral Reef Research and Monitoring Center.

A 10-min time transect with a 5-m width was employed using SCUBA at each site, and all individuals of damselfish species on the time transect were recorded using a data collection board. Prior to observations, a 5-m reference tape measure was laid on the sea floor. Then, an observer (A.N.) checked the visual estimates of 5 m width (2.5 m width in each side). A portable GPS receiver was used to measure the length of each time transect. The average distance covered was 155.1 m ± 25.3 m standard deviation (minimum length = 101 m; maximum length = 234 m). The fish density of each damselfish species per each site (number of individuals per 500 m2) was determined from the number of individual fish and the length of the 10-min transect. Depth profiles were recorded using a dive computer at a recording interval of 60 s. Ten depth values were obtained for each site (one depth point per minute × for 10 min). The obtained values were averaged for each site and used for subsequent analysis. The water depths ranged from 3.4 m to 11.0 m (7.65 m ± 1.89 m standard deviation).

During the observation period, 47 species and one genus (Stegastes) were identified at the study site, and 26 species were observed at high densities (Fig. S1). Given the similar morphological traits (body size and coloration) of Stegastes spp. as well as that the observations should be conducted while swimming, species-level identification for Stegastes spp. was not be conducted. Thus, Stegastes spp. was excluded from the analysis. Consequently, the above-mentioned 26 species were selected for further analysis because the 26 species comprised of 98.01% of the damselfish population at the study site (Fig. S1), and the use of dominant species would show more robust results about the spatial distribution in relation to environmental characteristics. Broad-scale distributions of each species were displayed by bubble plots on the study map, in which the bubble size represents the fish density per 500 m2.

Broad-scale substrate spatial variation

The substrate was recorded using a video camera (GoPro HERO5 Black), which was attached to the data collection board, from a top-down perspective along the transect at each site. Static images were extracted at 10-s intervals using QuickTime Player software (version 7.6) in the laboratory, yielding 61 static images for each site. The static images were used to calculate percentage coverage of each substrate. For each image, the substrate at the center of the static image was recorded. For example, if the number of points at a focal site was as “substrate A = 20, substrate B = 20, substrate C = 10, substrate D = 11”, the estimated coverage of each substrate at the focal site was calculated as “substrate A = 20/61 × 100 = 32.8%, substrate B = 20/61 × 100 = 32.8%, substrate C = 10/61 × 100 = 16.4%, substrate D = 21/61 × 100 = 18.0%”. The substrate was divided into 31 categories (Table S1): (1) staghorn Acropora, (2) branching Acropora, (3) bottlebrush Acropora, (4) corymbose Acropora, (5) tabular Acropora, (6) Pocillopora, (7) branching Isopora, (8) branching Millepora, (9) branching Porites, (10) other branching corals (other than genera Acropora, Isopora, Millepora and Porites), (11) foliose corals, (12) massive corals, (13) other live corals (encrusting corals and mushroom corals), (14) dead staghorn Acropora, (15) dead branching Acropora, (16) dead bottlebrush Acropora, (17) dead corymbose Acropora, (18) dead tabular Acropora, (19) dead Pocillopora, (20) dead branching Isopora, (21) dead branching Millepora, (22) dead branching Porites, (23) dead other branching corals, (24) dead foliose corals, (25) dead massive corals, (26) dead other corals, (27) soft corals, (28) rock (calcium carbonate substratum with lower substrate complexity than live corals), (29) coral rubble, (30) sand, and (31) macroalgae.

The data for substrate availability obtained from the 67 study sites were used for the further analysis (see “Data preparation for CCA” section).

Microhabitat-scale substrate association of fishes

Additional underwater observations were conducted at 19 sites from November 2021 to January 2024 using SCUBA to clarify the microhabitat-scale substrate associations of the 26 damselfish species (Fig. 1D). Four 20 m × 2 m transects were established at each site. Then, the substrate on which fish individuals were initially associated was recorded. In minimizing the impact on fish behaviors, an observer (A.N.) recorded fish data while pulling a 20-m tape measure. For each transect, the 20-m tape measure was set at the center of the transect. Substrates beneath the tape measure were recorded by using a video camera (GoPro HERO5 Black). Then, in the laboratory, substrate images were extracted at 10-cm intervals and divided into the above-mentioned 31 categories for analysis.

Analyses for broad-scale spatial distribution

For each fish species, a generalized linear model (GLM) was applied to examine the significant difference in fish density between exposed and inner reefs using R statistical computing language (R Core Team, 2022). The objective and explanatory variables were fish density and reef type (i.e., exposed reefs or inner reefs), respectively.

To consider the data distribution for broad-scale spatial distribution of each fish species, average and variance of the number of fish individuals at the 67 sites were calculated. This revealed that variance was greater than average (9.27-fold - 281.35-fold). Since Poisson distribution assumed that average and variance is almost equal, negative binomial distribution was applied for further data analysis.

The GLM was performed by “glm.nb” function of “MASS” package. The data were assumed to follow a negative binomial distribution with a log-link function. Considering that the fish count data at each site were obtained from a 10-min survey, the length of each time transect varied among the 67 sites. Thus, fish data were analyzed with the “offset” option in the R package using the length of each time transect.

After performing the GLM, the degree of zero-inflation was examined by using “check zeroinflation” function of “performance” package. This procedure revealed that two out of 26 species (Neopomacentrus anabatoides and Chromis ternatensis) showed a significant zero-inflation of data distribution under the assumption of negative binomial distribution. Thus, additional GLM was performed by “zeroinfl” function of “pscl” package for these two species with the assumption of zero-inflated negative binomial distribution.

The relationship between the broad-scale spatial distribution of the 26 damselfish species and the 32 environmental characteristics (31 substrate categories plus depth) was analyzed as follows: (1) detrended correspondence analysis (DCA) was performed to examine the species response (linear or unimodal) to the environmental characteristics using CANOCO software (Ter Braak & Smilauer, 2002); (2) since the DCA revealed the unimodal responses of species against environmental characteristics, canonical correspondence analysis (CCA) was performed to clarify the relationship. In addition, to identify the environmental characteristics that strongly affect the spatial distributions of the 26 damselfish species, forward selection was applied using CANOCO software.

Data preparation for CCA

Prior to the CCA, principal component analysis (PCA) was performed to reduce the number of independent variables, thereby avoiding multi-collinearity among the above-mentioned 32 environmental characteristics. The PCA was performed using PRIMER software (version 6). The PCA provided the principal component scores for 67 study sites along with five PC axes. Among the five PC axes, three axes (PC 1, PC 2 and PC 3) showed greater contributions to explain the overall trends in the site-specific difference in environmental characteristics (PC 1 = 64.1%; PC 2 = 18.4%; PC 3 = 7.2%). Thus, these principal scores were used as environmental variables for the CCA. For fish data, fish density data were log (x + 1)-transformed.

Analyses for microhabitat-scale substrate association

To clarify the overall trends in species-specific differences in substrate associations, cluster analysis using the group average linkage method with the Bray-Curtis similarity index was applied. Cluster analysis was performed using PRIMER software (version 6).

“Resource selection ratio” was applied to examine the substrate association (Manly et al., 2002) as:

wi=oi/πi

where wi is the resource selection probability function, oi is the proportion of the ith substrate that was used by a focal fish species, and πi is the proportion of the ith substrate that was available in the study area (Manly et al., 2002). For multiple comparisons, Bonferroni Z corrections (Quinn & Keough, 2002) was used to calculate the 95% confidence interval (CI) for each wi as:

95% CI=Za/2I [oi(1−oi)/(U+ πi2)]

where Za/2I is the critical value of the standard normal distribution corresponding to the upper tail area of a/2I, a is 0.05, I is the number of substrate categories, and U+ is the total number of individuals of the focal fish species. Substrates with wi ± 95% CI above and below one indicate a significantly positive and negative (non-positive, not avoidance) association, respectively. Substrates with wi ± 95% CI encompassing one showed no significant positive or negative association.

Degree of dependence on live or acroporid corals

In accordance with Pratchett et al. (2016), obligate coral dwellers are defined as species in which over 80% of the individuals are associated with live corals. In addition, the author of this study (A.N.) proposed two additional definitions for the degree of dependence on live corals as follows: (1) greater degree of dependence (over 60% of the individuals are associated with live corals) and (2) some extent of dependence (over 40% of the individuals are associated with live corals).

Dependence on acroporid corals is also regarded as an indicator of the susceptibility of damselfish species to the destruction of coral assemblages by mass coral bleaching events and outbreaks of crown-of-thorns starfish (Pratchett et al., 2008). The degree of dependence on live or acroporid corals was calculated as follows:

Degreeofdependenceonliveoracroporidcorals(%)=[(totalnumberoffishindividualsthatwereassociatedwithliveoracroporidcorals)/(totalnumberofobservedindividuals)]×100

This index was calculated for each damselfish species using microhabitat-scale substrate association data.

Analyses of habitat partitioning using niche overlap index

To examine both broad-scale and microhabitat-scale habitat partitioning, Pianka’s index (Pianka, 1973) was applied as follows:

Ojk=Okj=∑​pik•pik/[(∑​pij2•∑​pij2)]

where Ojk and Okj are the niche overlap indices between jth and kth species, respectively; pij and pik represent the proportions of the ith resource used by the jth and kth species, respectively. The values of this index range from 0 to 1 with greater values representing a greater niche overlap (habitat overlap) and vice versa. This index was calculated for both broad-scale spatial distribution and microhabitat-scale substrate associations. Then, the relationship between the broad-scale and microhabitat-scale niche overlap indices was plotted as a two-dimensional graph for a focal species to the other species (x-axis = broad-scale index, y-axis = microhabitat-scale index).

The setting value of the threshold whether habitat partitioning was found or not was 0.5 based on Harmácková, Remesová & Remes (2019), indicating that Ojk < 0.5 and Ojk > 0.5 represent greater degree of habitat partitioning and habitat overlap, respectively (Fig. S2).

Results

Overall trends in broad-scale spatial distribution

The map and GLM revealed the overall trends in the species-specific spatial distribution of the 26 fish species at the 67 study sites (Figs. 2–4, Table 1).

Figure 2 (A–J) Broad-scale spatial distributions of 10 damselfish species at 67 study sites, showing greater density in the exposed reef.

The circle diameter represents the density per 100 m × 5 m. Magenta and yellow symbols represent the sites in the exposed reefs and inner reefs, respectively. Aerial photographs were provided by the International Coral Reef Research and Monitoring Center. Photographs of all fish species were taken by the author (A. Nanami).

Figure 3 (A–J) Broad-scale spatial distributions of eight damselfish species at 67 study sites, showing no significant difference in fish density between exposed and inner reefs.

The circle diameter represents the density per 100 m × 5 m. Magenta and yellow symbols represent the sites in the exposed reefs and inner reefs, respectively.

Figure 4 (A–H) Broad-scale spatial distributions of eight damselfish species at 67 study site, showing significant greater density in the inner reef.

The circle diameter represents density per 100 m × 5 m. Magenta and yellow symbols represent the sites at exposed reefs and inner reefs, respectively. Aerial photographs were provided by the International Coral Reef Research and Monitoring Center. Photographs of all fish species were taken by the author (A. Nanami).

Table 1 Average density of 26 damselfish species at the 67 study sites in broad-scale survey and results of generalized liner model (GLM) to examine the significant difference in fish density between exposed reefs and inner reefs.

	Average density per 500 m2 ± standard deviation			
Fish species	Exposed reef (n = 31)	Inner reef (n = 36)	Results of GLM	p-value	
Abudefduf vaigiensis	3.9 ± 17.8	0	*	–	
Chromis atripes	5.7 ± 23.0	0	*	–	
Chromis vanderbilti	17.8 ± 44.5	0	*	–	
Chromis margaritifer	18.7 ± 25.2	0.4 ± 1.2	Exposed > Inner	< 0.001	
Chromis ovatiformis	1.8 ± 4.2	0.1 ± 0.3	Exposed > Inner	0.001	
Chrysiptera rex	13.6 ± 14.0	1.2 ± 3.5	Exposed > Inner	< 0.001	
Pomacentrus lepidogenys	48.4 ± 80.1	4.7 ± 11.9	Exposed > Inner	< 0.001	
Pomacentrus philippinus	25.2 ± 25.4	1.2 ± 2.6	Exposed > Inner	< 0.001	
Pomacentrus vaiuli	4.7 ± 4.8	1.8 ± 5.5	Exposed > Inner	0.038	
Pomachromis richardsoni	119.6 ± 156.2	6.6 ± 22.3	Exposed > Inner	< 0.001	
Amblyglyphidodon leucogaster	1.3 ± 3.9	1.7 ± 2.8	N.S.	0.680	
Chromis chrysura	1.8 ± 5.0	0.8 ± 4.0	N.S.	0.458	
Chromis viridis	8.9 ± 47.2	22.6 ± 47.3	N.S.	0.267	
Chrysiptera cyanea	0.5 ± 2.8	0.8 ± 2.3	N.S.	0.647	
Neopomacentrus anabatoides	8.9 ± 49.4	3.4 ± 17.4	N.S.	0.656	
Pomacentrus coelestis	0.8 ± 3.9	1.8 ± 6.7	N.S.	0.504	
Pomacentrus moluccensis	3.0 ± 8.1	6.7 ± 10.9	N.S.	0.119	
Pomacentrus sp.1	0.4 ± 1.7	1.2 ± 2.4	N.S.	0.135	
Amblyglyphidodon curacao	0.3 ± 0.9	4.7 ± 6.6	Inner > Exposed	< 0.001	
Chromis ternatensis	0.3 ± 1.5	0.8 ± 5.1	Inner > Exposed	0.003	
Dascyllus aruanus	0.8 ± 4.3	12.2 ± 18.3	Inner > Exposed	< 0.001	
Dascyllus reticulatus	0.4 ± 1.4	3.6 ± 7.3	Inner > Exposed	0.012	
Pomacentrus alexanderae	12.6 ± 31.0	52.1 ± 85.7	Inner > Exposed	0.026	
Pomacentrus amboinensis	1.9 ± 5.1	18.9 ± 19.6	Inner > Exposed	< 0.001	
Chrysiptera parasema	0	24.6 ± 55.1	**	–	
Pomacentrus sp.2	0	1.9 ± 4.8	**	–	
Note:

N.S., No-significant difference.

* : GLM was not performed due to no fish individuals at inner reefs.

** : GLM was not performed due to no fish individuals at exposed reefs.

Three species (Abudefduf vaigiensis, Chromis atripes and Chromis vanderbilti) were found only in the exposed reefs (Figs. 2A–2C, Table 1). Seven species (C. margaritifer, C. ovatiformis, Chrysiptera rex, Pomacentrus lepidogenys, P. philippinus, P. vaiuli and Pomachromis richardsoni) showed significantly greater densities in the exposed reefs (Figs. 2D–2J, Table 1; p < 0.05 for all seven species).

Eight species (Amblyglyphidodon leucogaster, Chromis chrysura, C. viridis, Chrysiptera cyanea, Neopomacentrus anabatoides, P. coelestis, P. moluccensis and P. sp. 1) showed no significant difference in density between the exposed reefs and inner reefs (Fig. 3, Table 1; p > 0.05).

By contrast, six species (Amblyglyphidodon curacao, Chromis ternatensis, Dascyllus aruanus, D. reticulatus, Pomacentrus alexanderae, P. amboinensis,) showed significantly greater densities in the inner reefs (Figs. 4A–4F, Table 1; p < 0.05 for all six species). Two species (Chrysiptera parasema and Pomacentrus sp. 2) were found only in the inner reefs (Figs. 4G, 4H, Table 1).

Broad-scale spatial distribution in relation to environmental characteristics

The results of PCA revealed the relationship between the 32 environmental characteristics and the three PC axes (Fig. S3). PC 1 represents greater coverage of coral rubble and sand on the positive axis, and greater coverage of rock on the negative axis. PC 2 represents greater coverage of sand on the positive axis, and greater coverage of live corals (e.g., branching Acropora, bottlebrush Acropora, branching Millepora and other corals), dead corals (dead branching Acropora and dead other corals) and coral rubble on the negative axis. PC 3 represents greater coverage of live corals (e.g., branching Acropora, bottlebrush Acropora, branching Millepora, massive corals and other corals), dead corals (dead branching Acropora and dead bottlebrush Acropora) and macroalgae on the positive axis, and greater coverage of coral rubble on the negative axis.

The CCA revealed that the fish assemblages were primarily divided into three groups in relation to the environmental characteristics (Fig. 5). The first group consisted of 11 species (Abudefduf vaigiensis, Chromis atripes, C. chrysura, C. vanderbilti, C. margaritifer, C. ovatiformis, Chrysiptera rex, Pomacentrus lepidogenys, P. philippinus, P. vaiuli and Pomachromis richardsoni) that were found at the first and fourth quadrants of the CCA plot (quadrants with the minus direction of PC axis 1), indicating that these species primarily occurred in the exposed reefs with greater coverage of rock (Figs. S3A, 5). The second group consisted of nine species (Amblyglyphidodon curacao, A. leucogaster, Chromis ternatensis, Chrysiptera parasema, Neopomacentrus anabatoides, Pomacentrus alexanderae, P. moluccensis, P. sp. 1 and P. sp. 2) that were found in the second quadrant of the CCA plot (quadrant with the plus directions of PC axes 1 and 3), indicating that these species primarily occurred in the inner reefs with greater coverage of live corals (branching Acropora, bottlebrush Acropora, staghorn Acropora and branching Millepora), dead corals (dead branching Acropora and dead bottlebrush Acropora) and macroalgae (Figs. S3A, S3C, 5). The third group consisted of six species (Chromis viridis, Chrysiptera cyanea, Dascyllus aruanus, D. reticulatus, Pomacentrus amboinensis and P. coelestis) that were found at the third quadrant of the CCA plot (quadrant with the plus directions of PC axes 1 and 2), indicating that these species primarily occurred in the inner reefs with greater coverage of sand (Figs. S3A, S3B, 5).

Figure 5 Results of the canonical correspondence analysis (CCA).

The figure shows the relationship between the spatial distribution of the 26 damselfish species and environmental characteristics (three principal component axes that were obtained by the principal component analysis: see Materials & Methods and Fig. S3). In (A), red letters represent the dominant substrate types of each quadrant, which were extracted by the PCA (see Fig. S3). In (B), fish genera are indicated by different colors (gray: Abudefduf, green: Amblyglyphidodon, sky-blue: Chromis, purple: Chrysiptera, magenta: Dascyllus, white: Neopomacentrus, orange: Pomacentrus, black: Pomachromis). In (C), magenta and yellow symbols represent 31 sites in the exposed reefs and 36 sites in the inner reefs, respectively. In (D-K), the names of fish species are indicated by abbreviations (Ab.vai: Abudefduf vaigiensis, Am.cur: Amblyglyphidodon curacao, Am.leu: Amblyglyphidodon leucogaster, C.atr: Chromis atripes, C.chr: Chromis chrysura, C.mar: Chromis margaritifer, C.ova: Chromis ovatiformis, C.ter: Chromis ternatensis, C.van: Chromis vanderbilti, C.vir: Chromis viridis, Ch.cya: Chrysiptera cyanea, Ch.par: Chrysiptera parasema, Ch.rex: Chrysiptera rex, D.aru: Dascyllus aruanus, D.ret: Dascyllus reticulatus N.ana: Neopomacentrus anabatoides, P.amb: Pomacentrus amboinensis, P.ale: Pomacentrus alexanderae, P.coe: Pomacentrus coelestis, P.lep: Pomacentrus lepidogenys, P.mol: Pomacentrus moluccensis, P.phi: Pomacentrus philippinus, P.vai: Pomacentrus vaiuli, P.sp1: Pomacentrus sp. 1, Psp.2: Pomacentrus sp. 2, Po.ric: Pomachromis richardsoni). %-values in the parentheses in CCA axes 1 and 2 represent the percent variance of species-environment relation.

Microhabitat-scale substrate association

Species-specific variations in microhabitat-scale substrate associations were observed (Figs. 6–9, S4–S7). The cluster analysis revealed that the 24 species could be divided into six groups, and the remaining two species (Pomacentrus amboinensis and Neopomacentrus anabatoides) had unique patterns in terms of substrate association (Fig. 10).

Figure 6 (A–F) Relative frequency (%) of the fish individuals for six of 12 species that were classified into “Group A” in the cluster analysis (Fig. 10), and primarily associated with rock.

Dark-blue and magenta bars represent the relative frequencies of fish and substrates, respectively. The numbers above the dark-blue bars represent the number of individuals on the focal substrate. The results for the 13 types of dead corals were pooled for ease of viewing. For details about fish associations with each category of dead corals, see Fig. S4. Horizontal dashed lines represent the division of the four groups of substrates (acroporid corals, non-acroporid corals, dead corals and other substrates). Blue letters represent the number of observed individuals, the degree of dependence on live corals and the degree of dependence on acroporid corals. Photographs of all fish species were taken by the author (A. Nanami).

Figure 7 (A–F) Relative frequency (%) of the fish individuals for other six of 12 species that were classified into “Group A” in the cluster analysis (Fig. 10), and primarily associated with rock.

Dark-blue and magenta bars represent the relative frequencies of fish and substrates, respectively. The numbers above the dark-blue bars represent the number of individuals on the focal substrate. The results for the 13 types of dead corals were pooled for ease of viewing. For details about fish associations with each category of dead corals, see Fig. S5. Horizontal dashed lines represent the division of the four groups of substrates (acroporid corals, non-acroporid corals, dead corals and other substrates). Blue letters represent the number of observed individuals, the degree of dependence on live corals and the degree of dependence on acroporid corals. Photographs of all fish species were taken by the author (A. Nanami).

Figure 8 (A–G) Relative frequency (%) of fish individuals for seven species that were classified into “Group B”, “Group C” and “Group D” in the cluster analysis (Fig. 10).

Dark-blue and magenta bars represent the relative frequencies of fish and substrates, respectively. The numbers above the dark-blue bars represent the number of individuals on the focal substrate. The results for the 13 types of dead corals were pooled for ease of viewing. For details about fish associations with each category of dead corals, see Fig. S6. Horizontal dashed lines represent the division of the four groups of substrates (acroporid corals, non-acroporid corals, dead corals and other substrates). Blue letters represent the number of observed individuals, the degree of dependence on live corals and the degree of dependence on acroporid corals. Photographs of all fish species were taken by the author (A. Nanami).

Figure 9 (A–G) Relative frequency (%) of fish individuals for seven species that were classified into “Group E”, “Group F” and “out group” in the cluster analysis (Fig. 10).

Dark-blue and magenta bars represent the relative frequencies of fish and substrates, respectively. The numbers above the dark-blue bars represent the number of individuals on the focal substrate. The results for the 13 types of dead corals were pooled for ease of viewing. For details about fish associations with each category of dead corals, see Fig. S7. Horizontal dashed lines represent the division of the four groups of substrates (acroporid corals, non-acroporid corals, dead corals and other substrates). Blue letters represent the number of observed individuals, the degree of dependence on live corals and the degree of dependence on acroporid corals. Photographs of all fish species were taken by the author (A. Nanami).

Figure 10 Dendrogram for the hierarchical clustering of the 26 damselfish species based on the similarity of microhabitat association (group-average linkage method using the Bray–Curtis similarity index).

Thirteen types of dead corals were pooled and represented in gray color for ease of viewing. Species names are shown by three different colored letters based on the three groups in the results of the canonical correspondence analysis (CCA, see Fig. 5).

Group A is comprised of 12 species (Abudefduf vaigiensis, Chromis atripes, C. vanderbilti, C. chrysura, C. margaritifer, C. ovatiformis, Chrysiptera rex, Pomacentrus coelestis, P. philippinus, P. vaiuli, P. sp. 1 and Pomachromis richardsoni) (Figs. 6, 7, 10) that showed a significant positive association with rock (p < 0.0016, Table 2 except for Pomacentrus sp. 1).

Table 2 Results of statistical significance of substrate association of the damselfish species that were classified into “Group A” by cluster analysis (see Fig. 10).

		Pomacentrus sp.1	Pomacentrus vaiuli	Chromis margaritifer	Chromis chrysura	Pomachromis richardsoni	Pomacentrus coelestis	Chrysiptera rex	Pomacentrus philippinus	Abudefduf vaigiensis	Chromis vanderbilti	Chromis atripes	Chromis ovatiformis	
Acroporid coral	Staghorn Acropora	–	Negative	Negative	–	Negative	–	–	Negative	–	–	Negative	–	
	Branching Acropora	–	N.S.	N.S.	–	N.S.	–	N.S.	Negative	–	–	–	–	
	Bottlebrush Acropora	N.S.	N.S.	–	–	–	–	Negative	–	–	–	–	–	
	Corymbose Acropora	N.S.	N.S.	Positive	–	–	N.S.	N.S.	Negative	–	–	–	–	
	Tabular Acropora	–	N.S.	N.S.	–	–	–	N.S.	–	–	–	–	–	
Non-acroporid coral	Pocillopora	–	N.S.	Positive	–	–	–	–	Positive	–	–	N.S.	–	
	Branching Isopora	N.S.	–	–	–	–	–	–	–	–	–	–	–	
	Branching Millepora	N.S.	N.S.	N.S.	N.S.	N.S.	–	Negative	N.S.	–	–	N.S.	N.S.	
	Branching Porites	N.S.	N.S.	Negative	–	–	–	–	–	–	–	–	–	
	Branching coral	–	–	–	–	N.S.	–	–	–	–	–	N.S.	–	
	Foliose coral	–	–	–	–	–	–	–	Negative	–	–	N.S.	–	
	Massive coral	N.S.	N.S.	N.S.	–	–	–	Negative	Negative	–	N.S.	–	–	
	Other coral	–	Negative	Negative	N.S.	–	–	Negative	Negative	–	–	–	–	
Dead acroporid coral	Dead staghorn Acropora	–	N.S.	Negative	–	N.S.	–	N.S.	Negative	–	–	–	–	
	Dead branching Acropora	N.S.	N.S.	Negative	–	–	–	N.S.	–	–	–	–	–	
	Dead bottlebrush Acropora	–	–	–	–	–	–	–	–	–	–	–	–	
	Dead corymbose Acropora	–	N.S.	N.S.	–	–	N.S.	–	–	–	–	–	–	
	Dead tabular Acropora	N.S.	–	–	–	–	–	N.S.	–	–	–	–	–	
Dead non-acroporid coral	Dead Pocillopora	–	–	–	–	–	–	–	–	–	–	–	–	
	Dead branching Isopora	–	N.S.	–	–	–	–	–	–	–	–	–	–	
	Dead branching Millepora	–	N.S.	–	–	–	–	–	–	–	–	–	–	
	Dead branching Porites	–	–	–	–	–	–	–	–	–	–	–	–	
	Dead branching coral	–	–	–	–	N.S.	–	–	–	–	–	–	–	
	Dead foliose coral	–	–	–	–	–	–	–	–	–	–	–	–	
	Dead massive coral	–	–	–	–	–	–	–	–	–	–	–	–	
	Dead other coral	–	–	–	–	–	–	–	–	–	–	–	–	
Other substrates	Soft coral	–	–	N.S.	–	–	–	N.S.	N.S.	–	–	–	–	
	Rock	N.S.	Positive	Positive	Positive	Positive	Positive	Positive	Positive	Positive	Positive	Positive	Positive	
	Coral rubble	N.S.	Negative	–	–	–	–	Negative	–	–	–	–	–	
	Sand	–	–	–	–	–	–	–	–	–	–	–	–	
	Macroalgae	–	–	–	–	–	–	–	–	–	–	–	–	
Note:

Statistical significances were calculated by resource selection ratio for 31 categories of substrates. The actual quantitative results are shown in Table S2.

Group B is comprised of two species (Chrysiptera cyanea and Pomacentrus sp. 2) (Figs. 8A, 8B, 10) that showed a significant positive association with coral rubble and negative association with rock (p < 0.0016, Table 3).

Table 3 Results of statistical significance of substrate association of the damselfish species that were respectively classified into “Group B”, “Group C” and “Group D” by cluster analysis (see Fig. 10).

		Pomacentrus sp.2	Chrysipitera cyanea	Dascyllus reticulatus	Dascyllus aruanus	Chromis ternatensis	Pomacentrus lepidogenys	Chrysiptera parasema	
Acroporid coral	Staghorn Acropora	–	–	N.S.	N.S.	N.S.	N.S.	Positive	
	Branching Acropora	–	Positive	N.S.	N.S.	Positive	Positive	Positive	
	Bottlebrush Acropora	N.S.	N.S.	–	–	–	N.S.	Positive	
	Corymbose Acropora	–	N.S.	Positive	Positive	–	N.S.	Positive	
	Tabular Acropora	–	–	–	–	–	N.S.	–	
Non-acroporid coral	Pocillopora	–	–	Positive	Positive	–	N.S.	Negative	
	Branching Isopora	–	Positive	Positive	N.S.	–	Negative	Positive	
	Branching Millepora	–	N.S.	–	N.S.	–	Positive	Positive	
	Branching Porites	–	–	–	N.S.	N.S.	Negative	Positive	
	Branching coral	–	–	–	N.S.	–	N.S.	–	
	Foliose coral	–	–	–	–	–	N.S.	Positive	
	Massive coral	–	–	–	–	–	N.S.	–	
	Other coral	–	–	Negative	–	–	Negative	N.S.	
Dead acroporid coral	Dead staghorn Acropora	Positive	–	N.S.	–	N.S.	Positive	Negative	
	Dead branching Acropora	–	–	–	–	–	Positive	N.S.	
	Dead bottlebrush Acropora	N.S.	–	–	–	–	–	Negative	
	Dead corymbose Acropora	–	–	–	–	–	Negative	Negative	
	Dead tabular Acropora	–	–	–	–	–	–	–	
Dead non-acroporid coral	Dead Pocillopora	–	–	–	–	–	–	–	
	Dead branching Isopora	–	–	–	–	–	N.S.	N.S.	
	Dead branching Millepora	–	–	–	–	–	Positive	N.S.	
	Dead branching Porites	–	–	–	–	–	Negative	–	
	Dead branching coral	–	–	–	–	–	–	–	
	Dead foliose coral	–	–	–	–	–	–	–	
	Dead massive coral	–	–	–	–	–	–	–	
	Dead other coral	–	–	–	–	–	–	–	
Other substrates	Soft coral	–	–	–	–	–	N.S.	–	
	Rock	Negative	Negative	–	–	–	Negative	–	
	Coral rubble	Positive	Positive	–	–	–	–	–	
	Sand	–	–	–	–	–	–	–	
	Macroalgae	–	–	–	–	–	–	–	
Note:

Statistical significances were calculated by resource selection ratio for 31 categories of substrates. The actual quantitative results are shown in Table S3.

Group C is comprised of two species (Dascyllus aruanus and D. reticulatus) (Figs. 8C, 8D, 10) that showed significant positive associations with corymbose Acropora and Pocillopora (p < 0.0016, Table 3).

Group D is comprised of three species (Chromis ternatensis, Pomacentrus lepidogenys and Chrysiptera parasema) (Figs. 8E–8G, 10) that showed a significant positive association with branching Acropora (p < 0.0016, Table 3).

Group E is comprised of three species (Chromis viridis, Pomacentrus amboinensis and P. alexanderae) (Figs. 9A–9C, 10) that showed significant positive associations with branching Acropora, branching Isopora and branching Porites (p < 0.0016, Table 4).

Table 4 Results of statistical significance of substrate association of the damselfish species that were respectively classified into “Group E”, “Group F” and "outgroup" by cluster analysis (see Fig. 10).

		Chromis viridis	Pomacentrus moluccensis	Pomacentrus alexanderae	Amblyglyphidodon curacao	Amblyglyphidodon leucogaster	Pomacentrus amboinensis	Neopomacentrus anabatoides	
Acroporid coral	Staghorn Acropora	Negative	N.S.	Positive	Positive	Positive	N.S.	N.S.	
	Branching Acropora	Positive	Positive	Positive	N.S.	Positive	N.S.	–	
	Bottlebrush Acropora	–	N.S.	N.S.	N.S.	N.S.	N.S.	–	
	Corymbose Acropora	Positive	Positive	Negative	–	–	N.S.	–	
	Tabular Acropora	–	–	–	–	–	–	–	
Non-acroporid coral	Pocillopora	Negative	Negative	–	–	N.S.	–	–	
	Branching Isopora	Positive	Positive	Positive	N.S.	N.S.	Negative	–	
	Branching Millepora	Negative	Positive	Positive	Positive	Positive	Positive	Negative	
	Branching Porites	Positive	Positive	Positive	Positive	N.S.	N.S.	Positive	
	Branching coral	–	N.S.	N.S.	–	–	N.S.	–	
	Foliose coral	Negative	N.S.	Positive	–	N.S.	–	Positive	
	Massive coral	–	Negative	Negative	–	–	Negative	–	
	Other coral	–	–	Negative	–	–	–	–	
Dead acroporid coral	Dead staghorn Acropora	–	–	Negative	N.S.	N.S.	–	–	
	Dead branching Acropora	–	Negative	Negative	–	–	–	–	
	Dead bottlebrush Acropora	–	Negative	N.S.	–	–	N.S.	–	
	Dead corymbose Acropora	–	N.S.	N.S.	N.S.	–	N.S.	–	
	Dead tabular Acropora	–	–	Negative	–	–	–	–	
Dead non-acroporid coral	Dead Pocillopora	–	–	–	–	–	–	–	
	Dead branching Isopora	–	N.S.	Positive	–	–	N.S.	–	
	Dead branching Millepora	–	N.S.	N.S.	–	–	–	–	
	Dead branching Porites	Negative	–	Positive	–	–	N.S.	Positive	
	Dead branching coral	–	Negative	–	–	–	–	–	
	Dead foliose coral	–	–	–	–	–	–	–	
	Dead massive coral	–	–	–	–	–	N.S.	–	
	Dead other coral	–	–	–	–	–	–	–	
Other substrates	Soft coral	–	–	–	–	–	–	–	
	Rock	–	Negative	Negative	Negative	Negative	Negative	Negative	
	Coral rubble	–	–	–	–	–	N.S.	–	
	Sand	–	–	–	–	–	–	–	
	Macroalgae	–	–	–	–	–	–	–	
Note:

Statistical significances were calculated by resource selection ratio for 31 categories of substrates. The actual quantitative results are shown in Table S4.

Group F is comprised of two species (Amblyglyphidodon curacao and Am. leucogaster) (Figs. 9D, 9E, 10) that showed positive associations with staghorn Acropora and branching Millepora, as well as a negative association with rock (p < 0.0016, Table 4).

For the remaining two species, Pomacentrus amboinensis showed a significant positive association with branching Millepora (p < 0.0016, Fig. 9F, Table 4), whereas Neopomacentrus anabatoides showed significant positive associations with branching Porites, foliose corals and dead branching Porites (p < 0.0016, Fig. 9G, Table 4).

Dependence on live or acroporid corals

Among the 26 species, nine species (Dascyllus aruanus, D. reticulatus, Chromis ternatensis, Chrysiptera parasema, C. viridis, Pomacentrus moluccensis, P. alexanderae, Amblyglyphidodon curacao, and Am. leucogaster) were categorized as obligate coral dwellers (>80% of the individuals are associated with live corals; Figs. 8C–8E, 8G, 9A–9E).

Three species (Chromis ternatensis, Chrysiptera parasema and Amblyglyphidodon leucogaster) showed a greater degree of dependence on acroporid corals (>60% individuals were associated with acroporid coral) (Figs. 8E, 8G, 9E). Another three species (Dascyllus aruanus, D. reticulatus and Pomacentrus lepidogenys) also showed some extent of dependence on acroporid corals (>40% individuals were associated with acroporid coral) (Figs. 8C, 8D, 8F).

By contrast, 10 species (Abudefduf vaigiensis, Chromis atripes, C. chrysura, C. ovatiformis, C. vanderbilti, Chrysiptera rex, Pomacentrus coelestis, P. philippinus, P. sp. 2 and Pomachromis richardsoni) showed a lower degree of dependence on live corals (less than 16% individuals were associated with live corals) and acroporid corals (less than 10% individuals were associated with acroporid coral).

Habitat partitioning among multiple species in two spatial scales

Fourteen species (Amblyglyphidodon curacao, Am. leucogaster, Chromis chrysura, C. viridis, Chrysiptera cyanea, Ch. parasema, Dascyllus aruanus, D. reticulatus, Neopomacentrus anabatoides, Pomacentrus amboinensis, P. coelestis, P. vaiuli, P. sp. 1 and P. sp. 2) showed relatively greater habitat partitioning (Ojk < 0.5) with other species at the broad-scale and/or microhabitat-scale (Fig. 11, Table S2).

Figure 11 (A–N) Relationship between broad-scale Pianka’s niche overlap index and microhabitat-scale index for 14 species that showing greater degree of habitat partitioning among the species.

The setting value of the threshold for habitat partitioning was found to be 0.5 based on Harmácková, Remesová & Remes (2019), showing that numerical value < 0.5 and > 0.5 represent greater degree of niche partitioning (habitat partitioning) and niche overlap (habitat overlap), respectively (see also Fig. S2). Photographs of all fish species were taken by the author (A. Nanami).

For the remaining 12 species, greater habitat overlaps were found within four fish groups, each consisting of three fish species (Fig. 12, Table S3): (1) Abudefduf vaigiensis, Chromis atripes and C. ovatiformis (Figs. 12A–12C, Table S3); (2) Chromis margaritifer, C. vanderbilti and Chrysiptera rex (Figs. 12D–12F, Table S3); (3) Chromis ternatensis, Pomacentrus alexanderae and P. moluccensis (Figs. 12G–12I, Table S3); and (4) Pomacentrus lepidogenys, P. philippinus and Pomachromis richardsoni (Figs. 12J–12L, Table S3). Chromis ovatiformis also showed habitat overlaps with Pomacentrus philippinus (Figs. 12C, 12K, Table S3).

Figure 12 (A–L) Relationship between the broad-scale Pianka’s niche overlap index and the microhabitat-scale index for 12 species that showing greater degree of habitat overlap among some species.

The setting value of the threshold for habitat partitioning was found to be 0.5 based on Harmácková, Remesová & Remes (2019), showing that numerical value < 0.5 and > 0.5 represent greater degree of niche partitioning (habitat partitioning) and niche overlap (habitat overlap), respectively (see also Fig. S2). Photographs of all fish species were taken by the author (A. Nanami).

Discussion

This study examined the spatial distribution and habitat partitioning of damselfishes on an Okinawan coral reef using two spatial scale approaches (broad-scale and microhabitat-scale). The results provide a more comprehensive understanding of the spatial distribution of damselfishes in relation to environmental characteristics, which can be applied for effective conservational planning.

Species distribution based on two spatial-scale approaches

Broad-scale analysis revealed that 11 species were primarily found in the exposed reefs with a greater coverage of rock. Among these, microhabitat-scale analysis revealed that 10 species (Abudefduf vaigiensis, Chromis atripes, C. chrysura, C. vanderbilti, C. margaritifer, C. ovatiformis, Chrysiptera rex, Pomacentrus philippinus, P. vaiuli and Pomachromis richardsoni) showed positive associations with rock. Since Froese & Pauly (2024) has shown that the fish length of these 10 species is less than 20 cm, small holes and fine-scale uneven surfaces of rock in the exposed reef can provide refuge space. Ticzon et al. (2012) and Nanami (2021) have suggested that a rocky surface inherently provides uneven surfaces and large holes, and it creates complex physical structures. The substrate complexity provided by rock increases the density of groupers and parrotfishes (Ticzon et al., 2012, Nanami, 2021), although the degree of complexity is lower than that of live corals. Thus, it is suggested that some damselfish species are associated with complex physical structures provided by non-coralline substrates as habitats and refuge spaces. One exception was Pomacentrus lepidogenys, which was positively associated with live corals (branching Acropora and branching Millepora) and dead corals (dead staghorn Acropora, dead branching Acropora and dead branching Millepora). This suggests that P. lepidogenys selectively utilized live and dead corals as habitat, although these substrates were not abundant in the exposed reef.

Broad-scale analysis also revealed that nine species were primarily found in the inner reef with greater coverage of live corals, dead corals and macroalgae. Microhabitat-scale analysis revealed that seven species (Amblyglyphidodon curacao, Am. leucogaster, Chromis ternatensis, Chrysiptera parasema, Neopomacentrus anabatoides, Pomacentrus alexanderae and P. moluccensis) showed positive associations with live corals (including both acroporid and non-acroporid corals) that have complex physical structures. Corals with complex structures form suitable refuge spaces for damselfishes (reviewed in Pratchett et al. (2016)), because such structures reduce fish mortality caused by predation (Almany, 2004). Two species (N. anabatoides and P. alexanderae) also showed positive associations with dead corals, suggesting that the complex structures provided by dead corals are utilized as habitats or refuge spaces to a certain extent. The exceptions were two species (Pomacentrus sp. 1 and sp. 2). Pomacentrus sp. 1 showed no significant association with any substrate in the microhabitat-scale approach, although this species was associated with a greater coverage of live corals, dead corals and macroalgae in broad-scale approach. Pomacentrus sp. 2 showed significant positive associations with dead staghorn Acropora and coral rubble. Although broad-scale analysis revealed that these nine species were primarily found in the inner reefs with a greater coverage of macroalgae, microhabitat-scale analysis showed that no individuals were associated with macroalgae. This suggests that these nine species primarily occur in the inner reefs with live and dead corals, where macroalgae also occur.

In addition, the broad-scale analysis revealed six species were associated with sites that have a greater coverage of sand, yet microhabitat-scale analysis revealed contrasting results in an association with coral and rock. Five species (Chromis viridis, Chrysiptera cyanea, Dascyllus aruanus, D. reticulatus and Pomacentrus amboinensis) and one species (P. coelestis) showed significant positive associations with live corals (including acroporid and non-acroporid corals) and rock, respectively. This finding suggests that these six species are associated with substrates that have complex physical structures, which are surrounded by sandy bottom. Considering that sandy bottom areas are primarily found in the inner reef, it is suggested that the six species prefer complex physical structure in the inner reefs.

These results suggest the importance of multi-scale approaches in examining the spatial distribution patterns of damselfishes. In particular, six species (Chromis viridis, Chrysiptera cyanea, Dascyllus aruanus, D. reticulatus, Pomacentrus amboinensis and P. coelestis) showed differences between broad-scale and microhabitat-scale analyses. Since damselfishes are closely associated with substrates with fine structures as habitats and as refuge spaces, broad-scale analysis might not detect precise aspects of substrate associations. By contrast, microhabitat-scale analysis may not detect landscape-level spatial distributions (exposed reefs or inner reefs). Thus, in addition to considering a broad-scale approach (e.g., exposed reefs vs. inner reefs), a microhabitat-scale approach (precise categorization of substrates) should be considered to examine the spatial distribution patterns in small-sized fishes with greater dependence on substrates.

Dependence on live corals

Nine of the 26 species (Dascyllus aruanus, D. reticulatus, Chromis ternatensis, C. viridis, Chrysiptera parasema, Pomacentrus moluccensis, P. alexanderae, Amblyglyphidodon curacao, Am. leucogaster) were categorized as obligate coral dwellers in the present study (>80% of the individuals are associated with live corals). The results were similar to the results of Pratchett et al. (2016), with exception of three species (Amblyglyphidodon curacao and Am. Leucogaster and Neopomacentrus anabatoides). In this study, two species (Amblyglyphidodon curacao and Am. leucogaster) and one species (Neopomacentrus anabatoides) were categorized as obligate and facultative coral dwellers, respectively. By contrast, Pratchett et al. (2016) categorized these species as facultative and obligate coral dwellers, respectively. This difference might be due to geographical variations in damselfish behavior and the species composition of live coral assemblages.

The degree of dependence on live corals provides some insights about the effects of coral assemblage degradation on damselfish assemblage structures. In particular, the dependence on acroporid corals might be an effective indicator for estimating the effects of mass coral bleaching events and outbreaks of crown-of-thorns starfish on the decline of damselfish populations (Pratchett et al., 2008, 2009; Coker, Wilson & Pratchett, 2014). The degree of dependence on acroporid corals for three species (Chromis ternatensis, Chrysiptera parasema and Amblyglyphidodon leucogaster) was over 60%, and the degree for the other three species (Dascyllus aruanus, D. reticulatus and Pomacentrus lepidogenys) was over 40%. These results suggest that the populations of these six species might be negatively impacted to some extent after the loss of acroporid corals. By contrast, since the degree of dependence on acroporid corals for the other 20 species was less than 40%, the negative impact on the population of the 20 species might be lower. For the remaining 10 species, these species might be more resilient to the loss of acroporid corals, since the degree of dependence on acroporid corals was less than 10% for the 10 species.

Some previous studies also showed the associations of fish with substrates that have less complex physical structures (Wilson et al., 2008). For example, some damselfish species (Chrysiptera rollandi, Dischistodus melanotus and Neoglyphidodon nigroris) showed significant positive associations with coral rubble in the Great Barrier Reef (Wilson et al., 2008). The results of the present study also showed that two species (Chrysiptera cyanea and Pomacentrus sp. 2) showed a significant positive association with coral rubble. This suggests that some damselfish species utilize the fine-scale space provided by coral rubble, and these species might have some degree of resilience to the degradation of live corals.

Habitat partitioning

Among the 26 species on the Okinawan coral reef, 14 species showed a greater degree of habitat partitioning at broad and/or microhabitat-scales. Several species showed differences in their spatial distribution between the exposed and inner reefs. For example, Chrysiptera cyanea and Pomacentrus sp. 2 showed similar patterns in microhabitat association but different patterns in the broad-scale spatial distribution. By contrast, several species (e.g., Amblyglyphidodon curacao and Pomacentrus sp. 2) showed different patterns in microhabitat associations among the species but no clear differences in broad-scale spatial distribution. These results suggest that both broad-scale environmental variation and microhabitat-scale substrate diversity provide diverse habitats at the present study site, supporting species coexistence via habitat partitioning.

By contrast, the remaining 12 species showed a certain degree of habitat overlap at the broad and/or microhabitat-scales. For example, Abudefduf vaigiensis and Chromis atripes showed greater habitat overlap at both scales (Pianka’s indices were over 0.97 in both spatial scales). The size variations in crevices and holes on the rocky surface might be the main factor supporting the coexistence, because the fish length of Abudefduf vaigiensis has been found to be greater than that of Chromis atripes (Froese & Pauly, 2024). More details about the precise aspects of the differences in architectural structures within rocky surfaces should be examined to explain such patterns of species coexistence. Another reason for the species coexistence is the prey item differences between the two species, as the former and latter species are a benthic organism feeder and a plankton feeder, respectively (Pratchett et al., 2016).

Chromis atripes also showed greater habitat overlap with C. ovatiformis, and both species are plankton feeders (Pratchett et al., 2016). However, Leray et al. (2018) showed prey item partitioning between two species of plankton feeders (Dascyllus flavicaudus and Chromis viridis) in the lagoon of Moorea, when the species composition of plankton assemblages in the digestive tract was precisely identified. Thus, precise identification of prey item plankton should be conducted, and prey item difference might be also found between the two species. All 12 species showed some niche overlaps, indicating similar broad-scale spatial distribution, microhabitat-scale substrate association or similar prey item categorization. However, as discussed above, more precise substrate identification and/or prey item identification might clarify niche partitioning among the 12 species in a more precise manner.

Other ecological processes (e.g., inter-specific competition, presence of con-specific individuals and stochastic processes of larval settlement) also promote species coexistence among damselfishes (Sweatman, 1983, 1985; Munday, Jones & Caley, 2001; Munday, 2004; Eurich, McCormick & Jones, 2018a, 2018b; Ebersole, 1985). These ecological processes might be among the factors that maintain the coexistence among the several species with greater habitat overlaps in this study. Comprehensive examinations including various ecological factors should be considered to explain the precise mechanisms that are responsible for the coexistence among damselfish species.

Marine protected areas to conserve damselfish species diversity

Various ecological aspects (e.g., diverse habitat) should be considered to establish effective marine protected areas (Kelleher, 1999; Green, White & Kilarski, 2013). This study revealed the species-specific broad-scale spatial distribution and microhabitat-scale habitat association of damselfishes in an Okinawan coral reef. Based on the results, several ecological factors should be incorporated to achieve effective MPAs to conserve damselfish species diversity as follows: (1) both exposed reefs and inner reefs should be protected; (2) diverse substrate types including various live coral species as well as non-coralline substrates should be protected; and (3) species-specific responses to habitat degradation should be more precisely clarified.

Conclusion

This study examined the broad-scale spatial distribution, microhabitat association, and habitat partitioning of damselfishes in the Okinawan coral reef. Broad-scale (exposed reefs vs. inner reefs) and microhabitat-scale aspects (coral morphology, live corals, dead corals and substrates with complex structures) affect the species-specific spatial distribution and substrate associations. These results suggest that habitat partitioning is one of the factors responsible for species coexistence in at least 14 species of the study species. Furthermore, the two spatial-scale viewpoints provide valuable insights into a more comprehensive understanding of the spatial distribution and species coexistence of damselfishes. These results provide valuable insights to establish effective MPAs in order to conserve damselfish species diversity.

Supplemental Information

Supplemental Information 1 Total number of individuals for each damselfish species during the survey period (July - December 2019).

Enclosed data in red square in (A) were shown as enlarged figure in (B) for easy-to-see. The blue dashed line in (B) represents the threshold between selected and unselected species for the analyses.

Supplemental Information 2 Schematic diagram showing the relationship between Pianka’s index for broad-scale spatial distribution and Pianka’s index for microhabitat-scale substrate association.

The setting value of threshold whether the habitat partitioning was found or not was 0.5 based on Harmácková, Remesová & Remes (2019). In this figure, the focal species shows habitat partitioning with species A and B for both spatial scales (niche partitioning for two spatial scales), species C and D for microhabitat scale (niche partitioning for microhabitat-scale), and species E and for broad-scale (niche partitioning for broad-scale). The focal species does not show habitat partitioning with species G and H (niche overlap for two spatial scales), suggesting that habitat partitioning is not the main factor that responsible for the coexistence with the focal species and species G, and with focal species and species H.

Supplemental Information 3 The results of the principal component analysis (PCA), explaining the relationship between the three principal component axes and 32 environmental variables that were applied to the canonical correspondence analysis (CCA).

Supplemental Information 4 Relative frequency (%) of fish individuals for 6 of 12 species that were classified into “Group A” in the cluster analysis (Fig. 9), and primarily associated with rock.

Numbers above bars represent the number of individuals on the focal substrate. Horizontal dashed lines represent the division of the five groups of substrates (acroporid corals, non-acroporid corals, dead acroporid corals, dead non-acroporid corals and other substrates). Blue letters represent the number of observed individuals, the degree of dependence on live corals and the degree of dependence on acroporid corals. Photographs of all fish species were taken by the author (A. Nanami).

Supplemental Information 5 Relative frequency (%) of the fish individuals for other 6 of 12 species that were classified into “Group A” in the cluster analysis (Fig. 9), and primarily associated with rock.

Numbers above bars represent the number of individuals on the focal substrate. Horizontal dashed lines represent the division of the five groups of substrates (acroporid corals, non-acroporid corals, dead acroporid corals, dead non-acroporid corals and other substrates). Blue letters represent the number of observed individuals, the degree of dependence on live corals and the degree of dependence on acroporid corals. Photographs of all fish species were taken by the author (A. Nanami).

Supplemental Information 6 Relative frequency (%) of fish individuals for 7 species that were classified into “Group B”, “Group C” and “Group D” in the cluster analysis (Fig. 9).

The numbers above bars represent the number of individuals on the focal substrate. Horizontal dashed lines represent the division of the five groups of substrates (acroporid corals, non-acroporid corals, dead acroporid corals, dead non-acroporid corals and other substrates). Blue letters represent the number of observed individuals, the degree of dependence on live corals and the degree of dependence on acroporid corals. Photographs of all fish species were taken by the author (A. Nanami).

Supplemental Information 7 Relative frequency (%) of fish individuals for 7 species that were classified into “Group E”, “Group F” and “out group” in the cluster analysis (Fig. 9).

The numbers above bars represent the number of individuals on the focal substrate. Horizontal dashed lines represent the division of the five groups of substrates (acroporid corals, non-acroporid corals, dead acroporid corals, dead non-acroporid corals and other substrates). Blue letters represent the number of observed individuals, the degree of dependence on live corals and the degree of dependence on acroporid corals. Photographs of all fish species were taken by the author (A. Nanami).

Supplemental Information 8 Thirty-one categories of substrate for the analyses.

Supplemental Information 9 Numerical values of broad-scale Pianka’s niche overlap index and microhabitat-scale index for 14 species that showing greater degree in habitat partitioning among the 26 species.

Supplemental Information 10 Numerical values of broad-scale Pianka’s niche overlap index and microhabitat-scale index for 12 species that showing greater degree in habitat overlaps among some species.

Orange cells represent greater overlaps for both broad-scale and microhabitat-scale.

Supplemental Information 11 Figure 2 raw data.

Supplemental Information 12 Figure 3 raw data.

Supplemental Information 13 Figure 4 raw data.

Supplemental Information 14 Figure 5 raw data.

Supplemental Information 15 Figure 6 raw data.

Supplemental Information 16 Figure 7 raw data.

Supplemental Information 17 Figure 8 raw data.

Supplemental Information 18 Figure 9 raw data.

Supplemental Information 19 Figure 10 raw data.

Supplemental Information 20 Figure 11 raw data.

Supplemental Information 21 Figure 12 raw data.

Supplemental Information 22 Table 1 raw data.

Supplemental Information 23 Table 2 raw data.

Supplemental Information 24 Table 3 raw data.

Supplemental Information 25 Table 4 raw data.

Supplemental Information 26 Figure S1 raw data.

Supplemental Information 27 Figure S3 raw data.

Supplemental Information 28 Figure S4 raw data.

Supplemental Information 29 Figure S5 raw data.

Supplemental Information 30 Figure S6 raw data.

Supplemental Information 31 Figure S7 raw data.

The author express grateful thanks to Masato Sunagawa and Sho Sunagawa for their field guide, Nobuo Motomiya, Kenta Oishi, Fumihiko Nakamura and Minoru Yoshida for field assistance, and the staff of Yaeyama Field Station of Fisheries Technology Institute for support during the present study. Constructive comments from two anonymous reviewers were much appreciated. The present study complies with the current laws in Japan.

Additional Information and Declarations

Competing Interests

The author declares that they have no competing interests.

Author Contributions

Atsushi Nanami conceived and designed the experiments, performed the experiments, analyzed the data, prepared figures and/or tables, authored or reviewed drafts of the article, and approved the final draft.

Animal Ethics

The following information was supplied relating to ethical approvals (i.e., approving body and any reference numbers):

I conducted underwater observations only and I did not collect any fish samples. In Japanese law, I do not have to get any permissions from any institutions.

Field Study Permissions

The following information was supplied relating to field study approvals (i.e., approving body and any reference numbers):

I conducted underwater observations only and I did not collect any fish samples. In Japanese law, I do not have to get any permissions from any institutions.

Data Availability

The following information was supplied regarding data availability:

The raw measurements are available in the Supplemental Files.

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
