# Peer review of "Broad-scale spatial distribution, microhabitat association and habitat partitioning of damselfishes (family Pomacentridae) on an Okinawan coral reef"

_PeerJ, doi:10.7717/peerj.18977_

## Round 0.1 · original submission · Major Revisions

· Academic Editor

Major Revisions

Thank you very much for your manuscript titled “Broad-scale spatial distribution, microhabitat association and habitat partitioning of damselfishes (family Pomacentridae) on an Okinawan coral reef” that you sent to PeerJ.

This study presents very valuable and relevant information because it analyses the large-scale and microhabitat (substrate)-level spatial distribution and habitat partitioning of 26 species of damselfish in a Japanese reef. The results obtained help to understand the spatial distribution of fishes as a function of substrate and the mechanisms of species coexistence in damselfish.

As you will see below, comments of the two referees suggest a major review before your paper can be published. Given this, I would like to see a major revision dealing with the comments. Their comments should provide a clear idea for you to review, hopefully improving the clarity and rigor of the presentation of your work. I will be happy to accept your article pending further revisions, detailed by the referees, which largely focus on introduction, methods, results and discussion. Also, the manuscript requires significant improvement in its writing.

Separate the analysis by groups of reptiles to understand possible patterns of distribution, richness and diversity. In addition to considering the distribution of the species or genera to interpret the results obtained. Also, some statistical points require more clarity. In discussion consider the ecology of the species and groups.

Please note that we consider these revisions to be important and your revised manuscript will likely need to be revised again.

Reviewer 1 ·

Basic reporting

The manuscript is presented well, with clear methods, results and figures, and presents interesting findings of cross-scale habitat partitioning in damselfish. The author identifies the value of cross-scale analysis in understanding species distributions, to which I agree, and I commend the authors attempt to address this knowledge gap. However, the manuscript requires significant improvement in its writing in terms of succinctness, clarity and grammar, before publication. There are areas where sentences repeat information (e.g. lines 96-99, 149-151). I have noted some areas of improvement below. However, I would suggest you have a colleague who is proficient in English and familiar with the subject matter review your manuscript and improve current phrasing to ensure that an international audience can clearly understand your text.

Abstract:
Your abstract requires much rewriting – there is too much detailed results (i.e. listing every species in each “group”). Instead, summarise your main results succinctly and provide some synthesis/meaning of the findings.

Introduction:
The differences and strengths of broad-scale vs microhabitat-scale analyses are clear and well described. However, I suggest you make clear in the first paragraph what knowledge gap your study is addressing. Emphasise that you examine both broad- and microhabitat-scale habitat associations together to explore more nuanced patterns.

Throughout the introduction, I do not understand your use of the word perspective, e.g. broad-scale perspective. Please alter to more appropriate phrasing, such as broad-scale patterns or broad-scale approaches.

Results:
I commend the author on how clearly they wrote the results of the cluster analysis, given the vast amount. However, the writing is rather repetitive and could be made even more succinct, such as “Group 1 comprised of 12 species (list of species) that were significantly associated with greater rock coverage” (lines 297-324) and so on.

The quantitative results must be included in your findings. For example, it is stated that the GLM revealed overall trends in species-specific spatial distribution, but no GLM results are given in brackets nor are in Table 1.

Discussion:
The opening of the discussion should highlight your main finding - that damselfish habitat partitioning occurs on both broad and microhabitat scales. Whereas, currently, the discussion jumps straight into more detailed findings.

The author makes a good attempt at exploring the impacts of their findings on habitat partitioning. However, this could be improved by synthesising more and writing their synthesis more clearly and succinctly.

Figures and Tables:
On the whole, your figures are clear and easy to understand. I particularly appreciated the use of photos of each species in the figures. However, there are a few points of improvement.

Figure 4: What do the percentages on the axis titles correspond to? Is it possible to include in the figure some representation of what the quadrants might mean? (e.g. a small rock/rubble picture in one corner, live coral in another).

Figures 5-8 are difficult to understand and not fully described. It is not clear what the blue and pink bars represent, although it seems that the pink bars are the same for all species? In the figure caption, you describe what the blue letters are but what does the frequency represent? Also, please place your x-axis title at the bottom as you have with your other figures.

Your tables do not include your actual quantitative results, just the phrases negative/positive/non-significant. You should include numbers where possible for transparency.

Line 66: what are typical examples?

Line 86: Please remove the word "recently", MPAs have been around for decades.

Lines 116-117: "spatial distributions of damselfishes by using integration with the multiple spatial scales have not yet been examined in this region" - absolutely, this should be at the start of your introduction to tell readers immediately what knowledge gap you are addressing.

Lines 149-151: Repetitive. Two sentences could be made into one, e.g. "The substrate was recorded using a video camera (give camera model) from a top-down perspective along the transects at each site."

Lines 268-276: Explanations of PC axes should be clearer and more succinct, such as PCA1 represents greater coverage of rubble and sand on the positive axis, and greater coverage of rock on the negative axis.

Line 329: It is unclear what is defined as “obligate coral dweller”. Please explain.

Lines 387-389: Write more succinctly, e.g. broad-scale analysis revealed six species were associated with a greater coverage of sand, yet microhabitat-scale analysis suggested contrasting results in an association with coral and rock.

Line 409-416: This paragraph is unclear. What species? What categorization? Suggest rewriting to make clear.

Line 421-426: Poor wording: “three species and other three species”. Also be clear what you mean by other 20 species.

Experimental design

The manuscript presents primary research that addresses a fundamental knowledge gap (though this should be made clearer, please see above comment). The study system and dataset is robust, and the analysis seems sound. However, improvements could be made in the justification of damselfish as a study system needs improvement. Whilst I agree that damselfish are an excellent study system for many research questions, including habitat partitioning, damselfish as a whole are increasing in abundance worldwide, therefore focussing on their conservation seems arbitrary. Instead, I suggest to focus on their abundant presence on coral reefs across the world, and how their niches overlap so may use habitat partitioning to coexist. Also their role as prey species, their role as an ecosystem engineer (if you are looking at territorial or farming species), etc.

Overall, the methods are described well and could be repeated. However, a few specific improvements should be made for clarification (see below). I thank you for providing the raw data, however a descriptive metadata file briefly describing your supplemental files would be useful to future readers. Additionally, tables do not include your actual quantitative results, just negative/positive/non-significant (see above comment)

Lines 144-145: More justification on why you only included 26 species, perhaps simply a percentage of how much of the damselfish population was made up of the other 21 species.

Lines 170-171: It is unclear whether you surveyed fish again, at the same time as the microhabitat-scale data collection (2021-2024), or whether you used the same data collected previously during broad-scale data collection (2019). If the latter, this must be clearly stated and justification given as to whether you can assume any associations found, when the data could be 5 years apart, are valid.

Validity of the findings

Conclusions link to original research question and are supported by the results. However, they should be written more clearly and succinctly, and your main finding should be made clear at the start of the discussion.

Please see above regarding missing statistical results.

Reviewer 2 ·

Basic reporting

1. The dependence on corals aspect in the results and discussion needs to be better set up in the introduction. Beyond a brief mention of specialists and generalists, I can’t find anything in the introduction that specifically pertains to this. Please give more detailed background on this, and maybe included it as part of your second aim in the final paragraph of the introduction.
2. In the introduction, while the overall background of the study is clear, the transition from discussing habitat in paragraph 3 to MPAs in paragraph 4, then back to habitat in paragraph 5 is a little jarring – I would consider switching the order of paragraphs 4 and 5. The mention of specialists and generalists in paragraph 3 can then be linked to habitat partitioning, and the background on MPAs will flow nicely into the mention of MPAs for damselfish conservation in paragraph 6. The discussion of MPAs also needs to be linked back to in the rest of the paper, for example in the discussion rather than saying that your results show the effects of coral loss, you could emphasise how your results show the importance of habitat diversity and how MPAs may help to preserve this.
3. It would be good to provide a short paragraph at the start of the discussion to outline the overall results and relevance of the study, before getting into the details
4. While the manuscript is overall well written, there are minor grammatical errors that in some areas make the writing less easy to understand (I have given some examples in the line-by-line feedback). I would recommend having the manuscript read over by a native English speaker to aid readability.
5. I would suggest removing species names from the abstract and just giving numbers of species categorised; this would allow you to fit more of your background and discussion into the abstract

Experimental design

Please clarify the following points in the methodology:
1. How was transect width measured?
2. Even though Stegastes weren’t included in the final analysis, it would be helpful to know why Stegastes were not identified to species level. In Supplementary figure 1, some species are given as “Pomacentrus sp. 1” etc. which I assume refers to unidentified species – how many Stegastes spp. were present?
3. Was location along the transect where static images were taken marked using the GPS receiver? Or was total transect length divided by 60 to find the length between images? When describing the PCA, you state one principal component score per site was produced, implying that one value per substrate per site was input. How was substrate prevalence at each site calculated?
4. Were microhabitat-scale associations also recorded on SCUBA? If transects were laid before fish location was recorded, is there a chance that fish were disturbed by laying transects and may have moved?
5. I understand that a Poisson distribution is often the logical choice for count data, but it would be helpful to know what, if any, data examination was carried out prior to assuming a Poisson distribution.
6. As I understand it, carrying out a PCA on 32 variables should give 32 components, but in line 198 you refer to three PC axes. Is this referring to the three components that captured the most variation? I see that the variation explained is reported in Fig. S3 – it would be helpful to either state the variation explained i.e., why you chose these three components, or to cite Fig. S3 here as well as in the results.

Validity of the findings

1. I’m not sure if the degree of dependence on live corals can be said to give quantitative estimates of the effects of coral degradation (Line 417). While it’s certainly possible to infer how species may be affected, calculating the degree of coral dependence is not precisely the same as showing quantitively how species would be affected if that coral was removed. Also, in the abstract and conclusion, you state that the species-specific manner of coral loss affects the damselfish population. While your results certainly show interactions between damselfish species and their habitat, your data does not specifically support any effects of coral loss, as you did not quantify coral loss. Please consider rephrasing.
2. While your findings are overall well-reported and interpreted, it would be helpful to provide actual values (possibly in the supplementary) for a) p values from the glms and b) resource selection ratios for cluster analyses.
3. In the “Dependence on live or acroporid corals” section, please report the level of coral dependence (presumably <40%?) of the remaining nine species.

Additional comments

This is a very interesting study and overall seems to have been well carried out. I have a few clarifications that I would like to see in the methods, and some suggestions for minor restructuring. My biggest concern is the implication in the abstract, discussion, and conclusion that the results can be extrapolated to directly show the effects of coral loss on damselfish populations. While you certainly show associations of species with specific habitats and can therefore speculate that the loss of said habitat would affect these species, your results don’t directly show the effects of coral loss on damselfish. I have highlighted a couple of areas where this is implied and that I think should therefore be rephrased. Additionally, in the introduction there is quite extensive discussion of marine protected areas, which isn’t really revisited in the rest of the paper. Possibly a better angle would be to discuss how your findings could be used to inform which habitats are key for damselfish and could therefore be considered when designating MPAs, though it’s important to emphasise that your results do not directly show how fish would be affected by coral loss. Otherwise, most of my comments are very minor and should be easy to address. I realise my review is quite extensive; this is intended to be constructive and shouldn’t be construed as criticism as I think this is overall a good manuscript.

Line-by-line feedback:

Introduction

Line 72: “several tens kilometers” should be changed to “several tens of kilometers”

Line 79: “several tens or several centimeters scale” is unclear – consider rephrasing to e.g. “a scale of several centimeters or decimeters”.

Line 80: Does “substrate forms” refer to the benthic substrate that may occur below corals e.g., sand or rock? This sentence is a little confusing. I would consider rephrasing to something like “…fine scale variables such as species of live corals and forms of non-coralline substrates”

Line 98-100: More detail is needed here. Is the degree of habitat partitioning great or small? An example of habitat partitioning in reef fishes would also be helpful for context here.

Line 102: “provide ecosystem services as ornamental fishes” is unclear to me – I would usually associate the phrase “ornamental fishes” with the trade in aquarium fish so I’m not sure how this links to ecosystem services.

Line 117: Remove “the” before “multiple spatial scales”

Line 118: “support” should be “supports”

Methods

Line 172: There seems to be word(s) missing here – what were substrates recorded with?

Line 211: Please provide a reference for the Bonferroni corrections

Line 221-225: Most of this should be in the introduction – but please keep the threshold values for obligate coral dependence in the methods

Line 242: “dimensional” is misspelled

Results

Line 268, 270 & throughout: Rephrasing “the plus direction” to “the positive direction” or “more positive values” would aid readability (also, in line 270, the words “the” and “plus” are the wrong way round). Likewise, “the minus direction” could be rephrased as e.g. “the negative direction”

Lines 296-320: Use “these” not “the” when referring to “X number of species” when reporting each result e.g. “Common characteristics of these two species”.

Line 307: rephrase as “…a significant positive association…”

Line 312: “comprised” is misspelled

Line 313-314: Rephrase as “A common characteristic of these three species was a significant positive association with branching Acropora”

Line 320: “a negative association”

Line 330-334: While you have stated in the methods that >80% of individuals found on Acropora indicates obligate dependence, the >60% and >40% threshold values also need to be stated and justified in the methods.

Line 332: “The other” should be “another”

Line 340, 342: It would be helpful to restate the 0.5 habitat overlap threshold value here e.g., “Fourteen species… showed relatively greater habitat partitioning (Ojk < 0.5)”

Line 342: I’m assuming the greater overlap occurs within the listed groups, but it would be helpful to state this more explicitly – even just changing “in” to “within” would make this clearer

Discussion

Line 357: The way this is phrased makes it sound like you measured the fish, when I’m assuming the size is in the citation given? Consider rephrasing.

Line 361: Remove “that”

Line 364: Remove “a”

Line 365: “which” is misspelled

Line 377: Reduce mortality from which sources?

Line 380-381: This sentence seems to contradict itself. Do you mean that there was no association with a broad-scale habitat, but association with microscale habitats?

Line 383: It would aid readability to restate that these are the nine species found primarily on the inner reef

Line 387: Not sure if “in contrast” is the best way to start this paragraph – it’s not really in contrast with the previously-described results

Line 411: Rephrase for clarity e.g. “three species were categorized differently here to in Pratchett et al. 2016”

Line 423: Remove “of”

Line 425: “than the other”

Line 429-430: “less than 10% [of] individuals were associated with acroporid coral” needs to be included in the results

Lines 446-447: You don’t need to restate the aim here

Line 451, 454: The sentences reading “This was the broad-scale/microhabitat-scale habitat partitioning” could be deleted or rephrased; at the moment, they are a bit redundant as you’ve just explained the result.

Line 455: Add the word “both” before broad-scale just for clarity

Line 458: Avoid starting paragraphs with “however”

Line 462: Saying that length “was greater” again implies that you measured yourself – this could be rephrased as e.g., “length has been found to be greater”

Line 465: “patterns”

Line 466: “a benthic organism feeder and a plankton feeder”

Line 478: “in a precise manner”

Liner 479: “In contrast” could be removed here – this doesn’t contrast with the results discussed in the previous paragraph

Line 482: “might be among the factors that maintain”
Line 485: “that are responsible”
Conclusion
Line 493-494: I would consider rephrasing the second half of this sentence for clarity, also because you can’t definitively state that habitat partitioning did not affect 12 of the species, just that you didn’t find any evidence of this. Could be rephrased to e.g., “evidence was found that habitat partitioning is one of the factors responsible for species coexistence in at least 14 of our study species”.

Line 495: “comprehensive understanding”

Line 496: “coexistence of”

Figures and Tables

Please include the meaning of the symbol colours in the Fig. 1 legend, as you have in Fig. 2

In Figs. 5-8, why are there blue and pink bars for the same species/substrate combination in some cases? Do the colours represent a different site/something else? Please clarify this in the figure legends.

In Fig. 4, it would be helpful to annotate the plots, or possibly just the first plot showing the PC axes, with the habitat variables indicated by each quarter. This would make interpreting the plot easier, without having to refer to Fig. S3

---

## Round 0.2 · Major Revisions

· Academic Editor

Major Revisions

After reviewing this revised version of your manuscript, I see that the main comments suggested by the reviewers have been included. However, there are still some details that need to be clarified before having a final version that can be published.

The introduction needs to be expanded by giving importance to the concept of habitat partitioning. Some points of methodology and data analysis also need to be clarified. The entire paper requires improvement in English writing.

Reviewer 1 ·

Basic reporting

The author has made improvements to their manuscript, most notably in their results section and abstract. However, I do not believe they have fully met the recommendations from the previous review. Whilst the research question, experimental design and manuscript are overall good and contribute to scientific understanding, there are areas for improvement before publication.

The manuscript requires improvement in its writing. I note that the author replied they used an English editing service, but there are still issues with succinctness, clarity and grammar, and I highly suggest a colleague proficient in English and familiar with the subject matter to review the manuscript and improve current phrasing.

You have made your first paragraph much more succinct and the aim of your paper, combining broad-scale and microhabitat-scale, is more clear. However, it should be made even clearer this is what you are doing. State "In this paper, I combine broad-scale and microhabitat-scale approaches to provide comprehensive understanding of these relationships." (line 68-70)

Experimental design

Whilst there has been some rewriting to justify the use of damselfishes, I still maintain that the emphasis on conserving these species is somewhat arbitrary. These species are increasing in abundance worldwide, actually to the detriment of coral reefs. I agree that they are a good group of fishes to explore habitat partitioning, given their vast diversity and the role they play on coral reefs, and believe this should be the focus. There is too much emphasis on MPA in the introduction, which is not much revisited in the rest of the paper. And although your findings are interesting and could be used to inform MPA positioning, it cannot be directly extrapolated. I recommend revising the manuscript with an emphasis on the importance of understanding habitat partitioning, and less emphasis on MPA which are not actually studied. The idea that your findings can be used to plan MPA can of course still be included, but must have less emphasis.

Lines 162-168: The justification on why you only included 26 species is still unclear. I suggest a percentage of how much of the damselfish population was made up of the other 21 species, to justify leaving them out. E.g. if the 21 species only made up 5% of the population, then you have accounted for 95%. The added justification of Stegastes spp throws up greater uncertainty. Did you exclude all Stegastes species? If so, I believe this to be an issue. Even if you include all Stegastes as one species, this could be more justifiable given their similar traits than leaving them out completely.

Validity of the findings

Poisson distributions for count data are generally the logical choice. However, there is no mention of data examination or model validation. Indeed, the data were "assumed" to follow Poisson (line 206). Data distributions and models must be validated. Poisson is not the only distribution for count data, e.g. non-binomial is also commonly used. Please validate your analysis and state how you have done so.

Additional comments

The design, analysis and interpretation of results in this manuscript are sound. However, I recommend revisions to the manuscript in terms of how the research and findings fit into our current understanding (see above for specific areas). I do believe this manuscript contributes to our understanding of habitat partitioning by combining different scale approaches and deserves to be published. Therefore, I hope these comments are seen as constructive and not criticism as I think this is overall a sound study.

Reviewer 2 ·

Basic reporting

The author has overall done a good job of addressing the comments, however there are a few points that still require clarification or editing before publication.

There are still many grammatical errors in the text, and I would strongly recommend having the manuscript proofread and edited by a native English speaker with knowledge of the field prior to publication. I don't want to be super pedantic about this but there are areas where the grammatical errors cause ambiguity and make the results/discussion more difficult to interpret e.g. Line 472-474, so I think the manuscript would really benefit from being proofread by an English speaker.

Experimental design

I'm still a bit unclear on how the values for each environmental characteristic were calculated, prior to carrying out the PCA. You state that photographs were taken at 61 points along the transect, and the substrate at the centre of each was recorded. Were the recordings then used to calculate percentage coverage for each substrate? Or area coverage of each substrate? Or just a count of photographs where each substrate was present? Basically, I'm asking what form the data input to the PCA took. This needs to be clarified in the methods.

My comment regarding data examination has not been adequately addressed - simply stating that the data is count data is not a strong enough basis to choose a Poisson test. My particular concern here is zero-inflation - often with count data particularly in wild populations there are a high number of zero values. If this is the case, a zero-inflated Poisson might be more appropriate. Your justification for using a Poisson needs to be clarified in the methods.

Validity of the findings

Overall, great job revising your conclusions, however the mention of coral loss in the conclusion is still present and I think should be removed (Line 541).

Additional comments

Line-by-line:

Line 103-104: The final sentence of this paragraph just seems like a restatement of the previous sentence and can probably be deleted and replaced with a more in-depth description of habitat partitioning and some examples of resource or habitat partitioning that allow species to coexist.

Line 252-255: You state "in this study [an] additional two definitions for the degree of dependence on line corals were proposed" - is this referring to your study or to Pratchett et al.? Please clarify.

Line 380-381: This is still a bit unclear. Consider rephrasing to "For the remaining 12 species, greater habitat overlaps were found within four groups, each consisting of three fish species"

Figure 2: Number of sites surveyed is missing from the figure legend

---

## Round 0.3 · Minor Revisions

· Academic Editor

Minor Revisions

After reviewing this revised version of your manuscript, I see that the main comments suggested by the reviewers have been included. However, there are still some details that need to be clarified before having a final version that can be published.

Reviewer 2 ·

Basic reporting

The author has effectively addressed all comments from my previous review. The manuscript is much improved, and I am now happy to recommend publication with a couple of very minor edits, described below.

Line 106: “Reef slope” is repeated twice

Experimental design

Lines 184-188: Thank you for clarifying the methods. Please add a sentence stating that the photographs were used to calculate percentage coverage of each substrate before describing how you calculated the percentages.

Validity of the findings

No comment

Additional comments

The data for all figures and tables would benefit from metadata descriptions included in the files.

---

## Round 0.4 · accepted · Accept

· Academic Editor

Accept

After reviewing this revised version of your manuscript, I see that the comments suggested by the reviewers have been included, therefore, I am satisfied with the current version and consider it ready for publication.